



# A comparison of PM$_{2.5}$-bound polycyclic aromatic hydrocarbons in summer
# Beijing (China) and Delhi (India)
Atallah. Elzein[1], Gareth J. Stewart[1], Stefan J. Swift[1], Beth S. Nelson[1], Leigh R. Crilley[2,6],
Mohammed S. Alam[2], Ernesto. Reyes-Villegas[3], Ranu. Gadi[4], Roy M. Harrison[2,7], Jacqueline
F. Hamilton[1], Alastair C. Lewis[5]
[1]Wolfson Atmospheric Chemistry Laboratories, Department of Chemistry, University of York, York, YO10 5DD,
United Kingdom.
[2]Division of Environmental Health & Risk Management, School of Geography, Earth & Environmental Sciences,
University of Birmingham, Birmingham, B15 2TT, United Kingdom.
[3]Department of Earth and Environmental Science, The University of Manchester, Manchester, M13 9PL, United
Kingdom.
[4] Indira Gandhi Delhi Technical University for Women, New Delhi, 110006, India.
[5]National Centre for Atmospheric Science, University of York, York, YO10 5DD, United Kingdom.
[6] Currently at: Department of Chemistry, York University, Toronto, ON, Canada.
[7] Also at: Department of Environmental Sciences / Center of Excellence in Environmental Studies, King
Abdulaziz University, PO Box 80203, Jeddah, 21589, Saudi Arabia.
*Correspondence to*: Atallah. Elzein (atallah.elzein@york.ac.uk)
**Abstract.**
Polycyclic aromatic hydrocarbons (PAHs) are ubiquitous pollutants in air, soil and water and known to
have harmful effects on human health and the environment. The diurnal and nocturnal variation of 17-
PAHs in ambient particle-bound PAHs were measured in urban Beijing (China) and Delhi (India)
during the summer season using GC-Q-TOF-MS. The mean concentration of particles less than 2.5
microns (PM$_{2.5}$) observed in Delhi was 3.6 times higher than in Beijing during the measurement period
in both the day-time and night-time. In Beijing, the mean concentration of the sum of the 17 PAHs
($\sum$17-PAHs) was $8.2 \pm 5.1$ ng m$^{-3}$ in daytime, with the highest contribution from Indeno[1,2,3-cd]pyrene
(12 %), while at night-time the total PAHs was $7.2 \pm 2.0$ ng m$^{-3}$, with the largest contribution from
Benzo[b]fluoranthene (14 %). In Delhi, the mean $\sum$17-PAHs was $13.6 \pm 5.9$ ng m$^{-3}$ in daytime, and
$22.7 \pm 9.4$ ng m$^{-3}$ at night-time, with the largest contribution from Indeno[1,2,3-cd]pyrene in both the
day (17 %) and night (20 %). Elevated mean concentrations of total PAHs in Delhi observed at night
were attributed to emissions from vehicles and biomass burning and to meteorological conditions
leading to their accumulation from a stable and low atmospheric boundary layer. Local emission sources
were typically identified as the major contributors to total measured PAHs, however, in Delhi 25 % of
the emissions were attributed to long-range atmospheric transport. Major emission sources were
characterized based on the contribution from each class of PAHs, with the 4, 5, and 6 ring PAHs
accounting ~ 95 % of the total PM$_{2.5}$-bound PAHs mass in both locations. The high contribution of 5
ring PAHs to total PAH concentration in summer Beijing and Delhi suggests a high contribution from
petroleum combustion. In Delhi, a high contribution from 6 ring PAHs was observed at night,
suggesting a potential emission source from the combustion of fuel and oil in power generators, widely
used in Delhi. The lifetime excess lung cancer risk (LECR) was calculated for Beijing and Delhi, with
the highest estimated risk attributed to Delhi (LECR = 155 per million people), 2.2 times higher than
Beijing risk assessment value (LECR = 70 per million people). Finally, we have assessed the emission
control policies in each city and identified those major sectors that could be subject to mitigation
measures.



## 1 Introduction

The significant increase of particulate matter (PM) and gaseous pollutants over the past decades in some rapidly expanding economies, has led to greater emphasis being placed on mitigation of emissions and management air quality health effects. To support such measures requires insight in both the sources of pollution, and the composition of pollution so that most harmful sources may be tackled as a priority. Although there have been recent improvements that have reduced primary particle concentrations in some regions, concentrations of many damaging gases and fine particles continue to exceed WHO guidelines (WHO, 2016), in megacities such as Beijing (Elzein et al., 2019; Lin et al., 2018; Gao and Ji., 2018) and Delhi (Kanawade et al., 2019; Sharma et al., 2007) and in many other cities around the world such as Cairo, Egypt (Cheng et al., 2016) and Islamabad, Pakistan (Mehmood et al., 2020). Growing populations, human activities, energy consumption and natural contributions (volcanic eruptions and forest fires) are an important contributor to particles emissions. PM monitoring and analysis become ever more important because of its adverse effect on human health. The chemical composition of airborne particles influences the health impacts, particularly the abundance of primary and secondary organic matter, metals, and ions (WHO, 2016; Bond et al., 2004; Saikawa et al., 2009). Fine particles become more harmful as particle size decreases (ultrafine), they enter the human body through the lungs and may translocate to other organs causing respiratory diseases and cancer (Schraufnagel., 2020). The greatest adverse effects on human health in epidemiological studies are currently associated with the mass of particles less than 2.5 microns in diameter ($PM_{2.5}$) (Raaschou-Nielsen et al., 2013, Pun et al., 2017, Hamra et al., 2014). The organic component of $PM_{2.5}$ consists of thousands of compounds, among them polycyclic aromatic hydrocarbons (PAHs), a particular class of species with high toxic potency. They are released into the atmosphere from both natural and anthropogenic sources. PAHs are considered ubiquitous in the environment and can be found in soil and water via dry or wet atmospheric deposition (Menzie et al., 1992, Meador et al., 1995). Their major emissions come from anthropogenic sources and include incomplete combustion of fossil fuels, vehicle exhaust emissions, cigarette emissions, agricultural burning and industrial activities (Saikawa et al., 2009). It has been shown that PAHs can react with atmospheric oxidants leading to the formation of secondary species with direct-acting mutagenicity and carcinogenicity and thus they can be significant contributors to the high toxicity of particles even at low PM levels (Nisbet and LaGoy 1992).

Beijing and Delhi often suffer from severe air pollution episodes, reaching high $PM_{2.5}$ concentrations and air quality index levels. The local government in Beijing has declared many different air quality actions since September 2013, resulting in a decrease in the concentration of total PAHs as reported in recent studies for the winter season in Beijing (Chen et al., 2017, Elzein et al., 2019, Feng et al., 2019). This has been attributed to the efforts made by the municipal government of Beijing to improve air quality and control emissions by reducing combustion sources and promoting the use of clean energy sources and electric vehicles.

Several anti-pollution measures have been introduced in Delhi in the last two decades such as, Bharat stage (equivalent to Euro standards), switching public transport from running on diesel to compressed natural gas (CNG), and applying "odd-even" vehicle number plate restriction during working days (Guttikunda et al., 2014; Goel and Guttikunda, 2015; Chowdhury et al., 2017). Despite the government effort to tackle air pollution in India and especially in Delhi, recent studies have showed that the air quality continue to be among the poorest in the world causing thousands of premature deaths (Tiwari et al., 2015, Ghude et al., 2016, Chowdhury and Dey., 2016, Pant et al., 2017, Conibear et al., 2018). PAHs emission sources in Delhi have previously been attributed to vehicle emissions, coal combustion, wood and burning leaves (Gadi et al., 2019; Shivani et al., 2019; Gupta et al., 2011; Sharma et al 2007). To the best of our knowledge, data on $PM_{2.5}$-bound PAHs in Delhi during the summer season (pre-monsoon: March – June) is scarce and limited to other periods of the year with low-time resolution ambient samples (sample averaging time of 24 h). It has been shown that concentrations of ambient





particle-bound PAH when collected over long sampling times are subject to higher uncertainties related
to sampling artefacts deriving from meteorological effects and oxidant concentrations such as ozone
(Goriaux et al., 2006; Tsapakis and Stephanou, 2003, 2007; Ringuet et al., 2012a; Keyte et al., 2013).
Using shorter time periods for ambient particle sampling (e.g. 3 and 4 h) has been suggested as offering
more accurate diagnosis of emission sources (Tian et al., 2017; Srivastava et al., 2018), shorter time
sampling is still scarce and limited to studies outside China and India (Reisen and Arey, 2004;
Srivastava et al., 2018). Considering the above and that particles collected during 24 hours sampling
time integrate both daytime and night-time chemistry together, we collected high frequency ambient air
particle samples ($PM_{2.5}$) in urban Beijing (China) and Delhi (India) to determine the temporal diurnal
and nocturnal variation of PAHs. A great advantage in this study is that all particle samples from both
campaigns were collected, extracted and analysed using the same analytical method, which provide
better comparison of the variation in PAHs between cities and on the feasibility and efficiency of
implementing emission control policies to improve air quality in both cities.
**2 Methods**
**2.1 Sampling campaigns**
Both measurement campaigns were part of the UK NERC / MRC Air Pollution and Human Health
(APHH) research programme. The sampling site in Beijing was located at the Institute of Atmospheric
Physics, Chinese Academy of Sciences in Beijing (39°58'28" N, 116°22'15" E) and the sampling site
in Delhi was located at Indira Gandhi Delhi Technical University for Women (28°39'52.6" N,
77°13'54.1" E). In both campaigns, the sampling equipment was installed on the roof of a 2-storey
building about 8 m above ground level. Prior to sampling the quartz microfiber filters (Whatman QM-
A, 20.3 × 25.4 cm, supplied by VWR U.K.) were baked at 550 °C for 5 h in order to eliminate any
organic matter. $PM_{2.5}$ filter samples were collected every 3 hours during daytime and over 15 h at night-
time, using a High-Volume Air Sampler (Ecotech HiVol 3000, Victoria, Australia) operating at 1.33
$m^3 min^{-1}$. The daytime sampling started at 8:30 in the morning and the filter was changed every 3 h.
Night-time sampling began at ~17:30 and ended at 08:30 the following day. Filters were collected for
20 days (22 May 2017 to 10 June 2017) totalling 80 filters during Beijing campaign, and for 9 days (28
May 2018 to 5 June 2018) totalling 35 filters during Delhi campaign. After sampling, filters were
wrapped in aluminium foil, sealed in polyethylene bags and stored at -20 °C until extraction and
analysis.
**2.2 Sample extraction**
Collected filters were cut using a cube cutter (1/16 of the filter) measuring a surface area equivalent to
24 $cm^2$. Each section was then cut into small pieces to fit inside 5 mL stainless steel extraction cells
used by a pressurized solvent extractor (Dionex, ASE 350). All samples were extracted in acetonitrile
(HPLC-grade) using the following method: Oven at 120°C, pressure at 1500 psi, rinse volume 60 %
and 60 s purge time for three consecutive 5 min cycles. The extraction time of each cell was about 25
min for a final volume of 20 mL. Prior to purification, extracts (V = 20 mL) were evaporated to
approximately 6 mL under a gentle stream of nitrogen. All samples and blanks were purified on solid
phase extraction (SPE) silica normal phase cartridge (1g/6mL; Sigma Aldrich) to reduce the impacts of
interfering compounds in the matrix and to help maintain a clean GC injection inlet liner. After the
purification step, the solution of each sample was evaporated to 1 mL under a gentle stream of nitrogen
at room temperature (20 °C) and transferred to a 1.5 mL autosampler amber vial. Each concentrated
sample was stored at 4 °C until analysis.



### 2.3 Analytical procedures

In this study, 17-PAHs were selected based on their presence within the particle phase and commercially available standards. These are listed in Table 1 and standards purchased from Sigma Aldrich, Alfa Aesar and Santa Cruz Biotechnology in the UK with a minimum purity of 97 %. In parallel to individual standards, a mixed solution of the 16 EPA PAHs (CRM47940, Supelco, Sigma Aldrich) of 10 µg ml$^{-1}$ in acetonitrile was also used. Standard solutions for calibrations were prepared in acetonitrile (HPLC grade, 99.9 % purity, Sigma Aldrich). Phenanthrene-d10 and pyrene-d10 were used as surrogate standards and were spiked over two blank filters and two sample filters from both campaigns, with concentration on filters corresponding to 300 ng (V = 60 µL from 5 ng µL$^{-1}$ in acetonitrile). Spiked filters solutions were analysed 10 times, and the average recovery efficiencies calculated from surrogate standards was 96 %, ranging from 88 % to 107 % for both compounds, phenanthrene-d10 and pyrene-d10. PAHs concentrations were corrected to the average recovery efficiencies. These two deuterated compounds were supplied by C/D/N isotopes and distributed by QMX Laboratories Ltd (Essex, UK). All PAH were quantified using a gas chromatography - time of flight - mass spectrometry system (GC Agilent 7890B coupled to an Agilent 7200 Q-TOF-MS). 1 µL of each sample was injected in pulsed splitless mode at 320 °C using an automated liquid injection with the GERSTEL MultiPurpose Sampler (MPS). Helium was used as carrier gas at 1.4 mL min$^{-1}$ and target compounds were eluted using the RXi-5ms (Restek GC column, Crossbond diphenyl dimethyl polysiloxane; length: 30 m, diameter: 0.25 mm, film thickness: 0.25 µm). The analysis time of each sample was set to 35 min using the following GC oven temperature programme: 65 °C for 4 min as a starting point and then increased to 185 °C at a heating rate of 40 °C min$^{-1}$ and held for 0.5 min, followed by a heating rate of 10 °C min$^{-1}$ to 240 °C and then ramped at 5 °C min$^{-1}$ until 320 °C and held isothermally for further 6 min to ensure all analytes eluted from the column. The MS was operated in Electron Ionisation (EI) mode at 70 eV with an emission current of 35 µA. 10-point calibration solutions were injected 4 times in the same sequence as for samples and covered the range from 1 pg µL$^{-1}$ to 1000 pg µL$^{-1}$, with a correlation coefficient from the linear regression between 0.970 and 0.999.

### 2.4 Error evaluation

As part of our method validation and in addition to recovery efficiency corrections, we have evaluated other possible factors that can affect our final true result. In this study, the solvent (acetonitrile) and field blanks (n = 3) were analysed following the same procedure as for the samples (Extraction, SPE, Evaporation) to determine any source of contamination during sample preparation and the analytical procedure. Whilst most target compounds were found to be below our limit of detection (S/N=3), three PAHs (Fluorene, Phenanthrene, Fluoranthene, and Pyrene) were quantified in field blanks and their contributions to the final data have been corrected.

To evaluate the agreement between repeated measurements (precision error), we have calculated the relative standard deviations (%RSD) from replicate analysis (n = 10) of two samples. The %RSD average of total PAHs was 8.7 % (range: 3.36 – 13.71 %) for Beijing samples and 4.2 % (range: 2.64 – 10.12 %) for Delhi samples. The %RSD average for deuterium labelled compounds spiked over two sample filters was ~ 3.6 % for both campaigns. The %RSD of each compound is shown in Table S1. The higher precision error (>10 %) attributed to a few compounds in Beijing samples (Table S1) is most probably related to the lower concentration of PAHs in the samples compared to Delhi samples and to samples analysed previously from Beijing in wintertime (Elzein et al., 2019). Moreover, the calibration offset and the influence of the sample matrix on the quantification step are an important source of systematic error and was estimated to be a maximum of 20 %. Therefore, the upper limit estimated error, combining the precision and the systematic effects, is 30 % for Beijing samples and 25 % for Delhi samples.





In addition, another type of error has been attributed to sampling artefacts. Previous studies (Schauer et
al., 2003, Goriaux et al., 2006, Tsapakis and Stephanou, 2003, Brown and Brown, 2012) have reported
a chemical decomposition of PAHs depending on the ambient concentration of ozone and sampling
time. Therefore, data from long sampling times and under very high ozone ambient concentrations (>70
ppb) may be biased by sampling artefacts of more than 100 % (Schauer et al., 2003, Goriaux et al.,
2006). However, at low ozone levels (< 30 ppb), negative artefacts were considered not significant
(Tsapakis and Stephanou 2003), whilst, at medium ozone levels (30-50 ppb) PAHs values were
underestimated by 30 % (Schauer et al., 2003). Tsapakis and Stephanou 2003, have reported a loss of
PAHs by 28 % due to ozone atmospheric concentration about 60 ppb and long sampling times of 24 h.
They have also suggested that long sampling time under low ozone concentration (< 30 ppb) do not
affect the concentration of collected PAHs in the gas or particle phases, while short sampling time (2
h) under ozone concentration about 60 ppb will reduce the concentration of PAHs by 17 %. In this
study, ozone concentrations were measured in both campaigns and averaged to the filter sampling time
to provide more accurate estimation on the negative sampling artefacts. The ozone concentration in
summer Beijing ranged between 3.7 and 140 ppb over the campaign (mean value: $56 \pm 31$ ppb),
approximately 5 times higher than in winter (mean value: $10.4 \pm 8.8$ ppb, Elzein et al., 2019). Daytime
ozone concentration ranged from 12 to 140 ppb (mean value: $63 \pm 30$ ppb), while over the night-time it
ranged from 4 to 74 ppb (mean value: $34 \pm 18$ ppb). Therefore, based on Tsapakis and Stephanou (2003)
study, the negative sampling artefacts due to ozone concentration was estimated to be 20 % for daytime
samples (3 h) and 10 % for the night-time samples (15 h) due to lower ozone concentration at night.
Using the same approach, daytime ozone concentration in Delhi ranged between 39 and 119 ppb, with
a mean value: $75 \pm 20$ ppb, while the night-time concentration ranged from 14 to 50 ppb (mean value:
$37 \pm 12$ ppb). Therefore, the estimation of the negative sampling artefacts on the data from Delhi ranged
between 15 and 30 %, with the highest error estimation attributable to daytime-time samples (3 h)
because of the higher ozone concentration during the day.
**3 Results and discussion**
**3.1 Concentration levels of $PM_{2.5}$ and Benzo[a]pyrene in summer Beijing and Delhi**
In 2016, the World Health Organisation (WHO, 2016) published an air quality guideline on outdoor air
pollution limits to help protect human health and reduce the risk of mortality due to fine particles. The
air quality standards for $PM_{2.5}$ were set by the ministry of environment in China and India, and published
in the WHO air quality guideline as an annual and 24 h mean concentration. The daily $PM_{2.5}$ (mean 24
h) guideline concentration is currently set at 75 and 60 $\mu g\ m^{-3}$ for China and India, respectively, while
the annual mean guideline concentration is currently set at 35 and 40 $\mu g\ m^{-3}$ for China and India,
respectively. $PM_{2.5}$ concentrations measured at the two sites were averaged to the filter sampling time
and are shown in Fig.1 and Fig. 2. In Beijing, the average 24 h $PM_{2.5}$ concentration was $39 \pm 21\ \mu g\ m^{-3}$
(range: 16 - 97 $\mu g\ m^{-3}$), exceeding the Chinese 24 h limit value (75 $\mu g\ m^{-3}$) on 1 day of the 20 sampling
days. The average daytime and night-time $PM_{2.5}$ concentrations in Beijing were $40 \pm 24\ \mu g\ m^{-3}$ and $38$
$\pm 24\ \mu g\ m^{-3}$, respectively.  Delhi showed an average 24 h $PM_{2.5}$ concentration of $143 \pm 27\ \mu g\ m^{-3}$ (range:
110 - 195 $\mu g\ m^{-3}$), exceeding the Indian 24 h limit value on all 9 sampling days. The average daytime
and night-time $PM_{2.5}$ concentrations in Delhi were $141 \pm 51\ \mu g\ m^{-3}$ and $140 \pm 26\ \mu g\ m^{-3}$, respectively.
$PM_{2.5}$ concentrations in Delhi have changed little in recent years; between 2008 and 2011 the daily
average of $PM_{2.5}$ concentrations was $123 \pm 87\ \mu g\ m^{-3}$ (Guttikunda and Calori., 2013), while the annual
average was reported to be $125.5 \pm 77.2\ \mu g\ m^{-3}$ between January 2013 and May 2014 (Winter: 196 $\mu g$
$m^{-3}$, Summer: 83.6 $\mu g\ m^{-3}$, Monsoon: 58.8 $\mu g\ m^{-3}$) (Sharma and Mandal., 2017). The limited change
seen in recent years may be associated with continued emissions from residential energy use, estimated
to contribute about 50 % of $PM_{2.5}$ airborne concentrations (Conibear et al., 2018, Butt et al., 2016). The





construction sector is fast growing in India, and the dust emitted from various activities (demolition,
excavation, drilling etc.) is also considered an important source of particles (Guttikunda et al., 2014).
The WHO has not published a guideline limit value for PAHs, but benzo[a]pyrene (BaP) is often used
as a marker of toxicity for all PAHs. The European Union has set an annual mean air quality limit of 1
ng m$^{-3}$ for BaP (WHO, 2016). The daily (24 h) concentration of BaP in summer Beijing (this study)
ranged from 0.49 to 1.18 ng m$^{-3}$ (average 0.80 ± 0.17 ng m$^{-3}$), about 19 times lower than previously
observed at this site in winter (Elzein et al., 2019). This is below the 24 h average limit value of 2.5 ng
m$^{-3}$, set in China by the Ministry of Ecology and Environment in 2012, on all of the 20 days of sampling
period. During the short summer measurement period in Delhi, BaP varied between 1.16 and 3.16 ng
m$^{-3}$ (average 1.78 ± 0.67 ng m$^{-3}$), and exceeded the threshold of 2.5 ng m$^{-3}$ on 1 day.

**3.2 Variability of PAHs in summer Beijing and Delhi**

A time series of the temporal variation of total PAHs in Beijing and Delhi are shown in Fig. 1 and Fig.
2, respectively. The box plots in Fig. 3 show a comparison of the measured concentrations of total 17-
PAHs between daytime and night-time in both cities.

**Beijing:**

The mean 3 h daytime concentration of $\sum$17-PAHs in Beijing was 8.2 ± 5.1 ng m$^{-3}$ ranging from 2.6 to
31.2 ng m$^{-3}$, while the mean 15 h night-time concentration was 7.2 ± 2.0 ng m$^{-3}$ ranging from 2.8 to 11.4
ng m$^{-3}$. The mean 24 h total concentration (combined results from daytime and night-time samples) of
the total 17-PAHs was 7.6 ± 1.9 ng m$^{-3}$ (range: 3.9 – 11.3 ng m$^{-3}$). This 24 h average is approximately
13 times lower than the average 24 h in winter time Beijing (97 ng m$^{-3}$) reported in our previous study
at the same location (Elzein et al., 2019). Similarly, previous studies in urban Beijing (Feng et al., 2005;
Wu et al., 2014; Gao and Ji., 2018; Song et al., 2019; Feng et al., 2019) have reported much lower
values of PAHs in summer than in winter, showing the important role of seasonal variation in
influencing ambient concentrations of PAHs. The dominant reason for this seasonal variation in Beijing
is the increase in energy consumption in winter and in particular the emissions from traditional
rural/urban heating methods using coal. The effect of dispersion in vertical and horizontal directions
and long-range transport due to air mass trajectory can also influence local PAHs concentration.
The photochemical effects on particulate PAHs between summer and winter are not clear in the
literature and were considered to play a minor role in seasonal variation of PAHs (Wu et al., 2014).
However, an important factor that might affect PAH levels is their degradation or transformation due
to high ozone ambient concentration level in summer, which were 5 times higher than in winter (Elzein
et al., 2019). This factor can negatively bias PAHs levels during filter sampling by more than 100 %
(discussed above in section 2.4). The gas phase concentrations of PAHs were not measured in this study,
but the distribution of PAHs between the gaseous and particulate phases is an important factor affecting
their fate in the environment (Lohmann and Lammel, 2004). The distribution of highly volatile PAHs
(e.g. 2-3 ring) are known to be influenced by temperature (Tsapakis and Stephanou., 2005; Gaga and
Ari., 2011; Verma et al., 2017), while the low volatile PAHs and PAHs-derivatives (e.g. oxygenated
and nitrated-PAHs) are mostly associated to the particle phase (Albinet et al., 2008; Liu et al., 2017,
Zhang et al., 2018). The spatial variation within urban Beijing might have little difference on pollutants
concentrations (He et al., 2001), the variation will increase when moving to suburban areas because it
consists of a large number of factories, airports, and power plant stations. Feng et al. (2005) compared
the total PAHs concentration in PM$_{2.5}$ at urban and suburban sites in Beijing at the same period of the
year (July and November 2002), and they reported higher values at the suburban site than at the urban
site by a factor ~1.5 in summer and ~ 2 in winter. The sampling location in this study is an urban area
surrounded by busy roads, residential buildings, an underground railway, and restaurants. Therefore, it





may be representative of the urban area of Beijing covering approximately half of the population in
Beijing metropolitan.
In Figure 1, the concentration of total PAHs in the first 3 h filter (08:30–11:30) of the day appear to be
higher than the rest of the day. These elevated concentrations are potentially associated with the early
morning rush hour time and vehicle emissions. The average night-time total PAHs concentration (7.2
ng m$^{-3}$) was in the range of the second and third "3 h" daytime average values, 7.8 and 6.4 ng m$^{-3}$
respectively. This indicates that the elevated total PAHs concentration in the first 3 h filter (08:30–
11:30) of the day is not related to accumulation of air pollutants at night-time but most likely related to
direct emissions from traffic in the early morning and particularly petrol combustion emissions (see
section 3.3).
To the best of our knowledge, previous studies in Beijing have not reported higher time resolution data
on total PAHs concentration, but mainly focused on the 24 h average concentration. A previous study
from Feng et al., (2005) reported a 24 h mean value of 25 ± 6.8 ng m$^{-3}$ in summer Beijing (July 2002),
approximately 3.3 times higher than our 24 h mean value (7.6 ng m$^{-3}$). A more recent study from Feng
et al., (2019) reported a 24 h mean value of 11 ± 5.9 ng m$^{-3}$ in Beijing in warm months (April to June
2015), 1.4 times higher than our 24 h mean value. In both studies of Feng et al., (2005 and 2019), the
urban sampling site was located at the campus of Peking University health science centre, a short
distance from our sampling site (~1 mile). In the same year of 2015 (July to September), Chen et al.,
(2017) and Zhang et al., (2020) reported the values of 9.7 ng m$^{-3}$ and 8.9 ng m$^{-3}$, respectively, for total
PAHs concentration in PM$_{2.5}$. In the study of Chen et al., (2017), the PM$_{2.5}$ samples were collected at
the campus of Beihang university (~2.5 miles from our sampling site). Zhang et al., (2020) sampling
site was located at a primary school in the Haidian district (~3.5 miles from our sampling site).
Furthermore, Gao and Ji (2018) reported 14.5 ± 1.3 ng m$^{-3}$ for total PAHs concentration in PM$_{2.5}$,
collected in summer Beijing (May-July, 2016) in the Haidian district (~4 miles from our sampling site).
Accordingly, the majority of previous studies have focused on studying PAHs in the Haidian district
(Wu et al., 2014; Chen et al., 2017; Gao and Ji, 2018; Feng et al., 2019; Zhang et al., 2020) because of
its high population density. The variation of PAHs concentration from different site locations in the
Haidian district may suggest that the spatial variation of PAHs in this area is not significant. Future
studies in different districts and rural areas of the metropolitan of Beijing would be helpful for
comparison of population exposures and spatial variation.
The results from studies of PAHs in summer Beijing in recent years (2015-2020) (Chen et al., 2017,
Gao and Ji., 2018, Feng et al., 2019, Zhang et al., 2020) show a continuous decrease in PAH
concentrations (range: 7 – 15 ng m$^{-3}$) in comparison with the previous decade (2000-2010; range: 11 –
31 ng m$^{-3}$) (Feng et al., 2005, Li et al., 2013, Wu et., 2014). This improvement in air quality (mitigating
the emissions of PAHs) could be related to meteorological conditions (e.g. temperature, boundary layer
height), but also to the anti-pollution actions adopted by the municipal government of Beijing in 2013
to continue tackling air pollution, by reducing combustion sources in the intervening years and
promoting the use of clean energy sources such as solar hot water heating systems, banning heavy duty
vehicles from circulating in daytime, public transport modernisation, promoting electric vehicles and
electric motorbikes.

**327  Delhi:**

The population in Delhi is projected to continue growing and to become the most populous city in the
world with 39 million people in 2030 (United Nations, 2019), living within a geographic area of 1483
km$^2$, of which 783 km$^2$ is designated as rural and 700 km$^2$ as urban (Nagar et al., 2014). The summer or
pre-monsoon season (March-June) has very high temperatures and low precipitation. The city is
surrounded by different climatic zones; the Thar desert in the west, the central hot plains to the south,
the Himalayas to the north, and the Indo Gangetic plain in the east (Nagar et al., 2014). The northern





and Eastern part of India are considered to be the most polluted part of the country (Guttikunda et al.,
335  2014).
Similar to Beijing, previous studies in Delhi mainly focused on the 24 h average concentration of total
PAHs. They have addressed the problem of air pollution across India, focusing on $PM_{2.5}$ trends, $PM_{2.5}$
health impact, and source apportionment (Chowdhury et al., 2007; Gupta et al., 2011; Chowdhury and
Dey., 2016; Pant et al., 2017; Chen et al., 2020), however, PAHs emissions and variation between
daytime and night-time have not been measured or discussed previously, and only few studies (Sharma
et al., 2007; Singh et al., 2011, Gadi et al., 2019) reported the 24 h mean concentration of $\sum$n-PAHs (n
> 10) in Delhi itself. In this study, the mean 24 h (combined results from daytime and night-time
samples) of the 17-PAHs was $19.3 \pm 7.1$ ng m$^{-3}$ ranging from 13.3 to 35 ng m$^{-3}$. Previous studies showed
a large spatial variation in PAHs concentrations within Delhi; Sharma et al., (2007) reported a mean 24
h value of $624.04 \pm 376.48$ ng m$^{-3}$ (~ 32 times higher than our mean value) for $\sum$12-PAHs at the South
of Delhi during the summer season of 2003. Singh et al., 2011 reported a mean 24 h value of $45.8 \pm$
$22.1$ ng m$^{-3}$ (~ 2.4 times higher than our mean value) for $\sum$16-PAHs at the East of Delhi during the
summer season of 2008. Gadi et al., (2019) reported an annual average of $277 \pm 126$ ng m$^{-3}$ for the
summation of 16-PAHs between December 2016 and December 2017, and $260 \pm 111$ ng m$^{-3}$ for the
summer season of 2017 (~ 13.5 times higher than our mean value, and ~ 5.7 times higher than Singh et
al., 2011 mean value). In the study of Gadi et al., (2019), the sampling site was in the same campus as
this study (Indira Gandhi Delhi Technical University for Women), the low $\sum$PAHs value in this study
is most probably due to the short summer measurement period in Delhi (9 days, 35 samples), coupled
with less pollution episodes for PAHs. It appears that the variation of PAHs concentration in Delhi
could be affected by multiple factors including the spatial variation, the input from multiple local
sources, the surroundings of the sampling site, and the meteorological conditions in the north of Delhi.
Therefore, the concentrations of ambient particle-bound PAH over longer averaging period such as the
24 h are subject to high uncertainty related to the multiple factors cited above. Higher frequency filter
sampling during 24 h can provide a better assessment of PAH concentrations and is more likely to
reflect direct source emission signals as modified by meteorology. This perspective also applies to other
Indian cities in future studies.
The mean 3 h $\sum$PAHs concentration in daytime samples in Delhi was $13.6 \pm 5.9$ ng m$^{-3}$ (~1.7 higher
than in Beijing) ranging from 8.4 to 36.6 ng m$^{-3}$, and the mean 15 h night-time samples was $22.7 \pm 9.4$
ng m$^{-3}$ (~3.2 higher than in Beijing) ranging from 13.8 to 42.9 ng m$^{-3}$ as shown in Figure 3.  During the
daytime, the total PAH concentrations were generally highest during the first filter sample (8:30 to
11:30 am) (Fig. 2), following the same trend as for Beijing suggesting vehicle emissions as a dominant
source. The mean total PAH concentration at night-time was ~ 1.7 times higher than the mean in
daytime. Higher total PAHs concentration at night could be related to emissions from biomass burning,
waste burning, solid fuel cooking and heavy duty diesels entering the city at night-time. The higher
PAHs concentration at night may also be attributed to the relatively lower temperature at night (~10 °C)
and lower atmospheric mixing heights (Fig. 4), weaker turbulence leading to lower pollutant dispersion
rates and absence of photodecomposition. Moreover, power cuts in India are frequent (Guttikunda et
al.,2014), especially when high demand occurs on air conditioners in summer (Harish et al., 2020),
which require in situ electricity generation using oil, diesel and petrol (Guttikunda et al.,2014). This
may be considered as an additional source of air pollution in a megacity like Delhi. $PM_{2.5}$ concentrations
increased on most nights in comparison with the preceding daytime sample. Residential energy use
across India has been reported to be an important source of $PM_{2.5}$ emissions, contributing 62 % in
summer and 70 % in winter of anthropogenic emissions of $PM_{2.5}$ (Conibear et al., 2018). This was also
confirmed in the study of Butt et al., (2016), showing that the impact of residential combustion
emissions on atmospheric aerosol across India is very important, accounting for 63 % of anthropogenic
black carbon and 78 % of anthropogenic particle organic matter emissions.



The mean 24 h values of ∑PAHs from this study ($19.3 \pm 7.1$ ng m$^{-3}$) and recent previous studies ($45.8$
$\pm 22.1$ ng m$^{-3}$ (Singh et al., 2011) and $260 \pm 111$ ng m$^{-3}$ (Gadi et al., 2019)) are high. This suggests the
need for the implementation of a residential emission control strategy through potentially more effective
alternative technologies such as the use of non-fossil fuel (biofuel) and clean energy sources (solar,
wind, hydro-electric power, natural gas) for domestic use, end the use of in situ power generators by
increasing electricity supply and load, and cutting emissions from open waste burning by implementing
efficient collection and disposal of waste.

**3.3 Major PAH, particle aging and traffic emissions**
In both campaigns, a high proportion of 5 and 6-ring PAHs were found in the particle phase (Table 1).
In Beijing, Benzo[b]fluoranthene, Benzo[k]fluoranthene, Benzo[a]pyrene and Indeno[1,2,3-cd]pyrene
were the four most abundant particle-bound PAHs in daytime samples, while Benzo[b]fluoranthene,
Indeno[1,2,3-cd]pyrene, and Benzo[ghi]perylene were the three dominant particle-bound PAHs in
night-time samples. Six major compounds were found in Delhi day and night samples,
Benzo[b]fluoranthene, Benzo[k]fluoranthene, Benzo[a]pyrene (BaP), Benzo[e]pyrene (BeP),
Indeno[1,2,3-cd]pyrene, and Benzo[ghi]perylene; the mean concentration of these compounds were
higher at night. Generally, the 2 and 3-ring PAHs are predominantly found in the gas phase and the 4-
ring PAHs partition between the gas and particle phase (Liu et al., 2013, Zhang et al., 2018). The study
from Liu et al., (2013) suggested that ambient temperature largely controls the gas–particle partitioning
of 2, 3, and 4-ring PAHs concentration in the gas and particle phases.
A number of molecular diagnostic ratios for source apportionment have been proposed in literature and
are still under debate (Larsen and Baker., 2003; Katsoyiannis et al., 2011; Keyte et al., 2013; Zheng et
al., 2017). They were considered uncertain in some studies because the results were not consistent and
reproducible and did not reflect known differences in sources in space and time unless the source is
very strong and the sampling measurements are made close to the known source. In addition, they may
be biased by atmospheric reactions and selective loss processes (Katsoyiannis et al., 2011; Zheng et al.,
2017). Among the PAHs, BaP is known to be a particularly carcinogenic compound inducing a
mutagenic effect in experimental animals and is used as key PAH marker of total exposure (WHO,
2016, IARC, 2012). BaP is mostly emitted from coal and biomass burning, and vehicle exhaust
emissions (Harrison et al., 1996; WHO, 2016). The sampling site in both Beijing and Delhi was at a
short distance from busy roads with significant vehicle exhaust emissions. BaP lifetime is affected by
light and oxidants in the atmosphere, BaP is far more reactive than its isomer BeP (Butler and Crossley,
1981, Ringuet et al., 2012b), thus the decline of the ratio BaP/BaP+BeP can be regarded as an indicator
of particle aging (Liu et al., 2013; Watson et al., 2016). Considering the above, BaP/BaP+BeP has been
used in this study to assess the contribution of local sources on particle composition. Generally, there
is no defined threshold value for BaP/ BaP+BeP which can distinguish aged particles from freshly
emitted. However, Watson et al., (2016) suggested that similar amounts of BaP and BeP [ratio = 0.5]
indicates that particles are freshly emitted and mostly affected by local emissions rather than long range
atmospheric transport. Moreover, Liu et al., (2013) compared BaP and BeP ratios at remote sites and
emission source regions and reported BaP/BeP lower than 0.4 means aged aerosol type, while a value
higher than 0.4 indicates local emission source. In this study, a ratio of BaP/BaP+BeP higher than 0.5
has been considered to characterize local emission sources. The ratio of BaP/(BaP+BeP) has been
calculated for both campaigns and results are shown in Figure 5. This ratio mainly varied between 0.5
and 0.6 in Beijing, indicating a dominant contribution from local sources. In Delhi, 25 % of the data
were below 0.5 indicating that aged particles might contribute to the air masses collected at the sampling
site but local emissions of PAH are still a significant source.
Since the influence from traffic emissions is very strong in both Beijing and Delhi, related diagnostic
ratios were used to distinguish between petrol and diesel, such as, Fluorene/Fluorene+Pyrene (< 0.5 for



Petrol engines and > 0.5 for diesel engines) and Pyrene/BaP (Ravindra et al., 2008; Tobiszewski and
Namiesnik., 2012; Watson et al., 2016; Zheng et al., 2017). Rogge et al., (1993) have quantified more
than 100 organic compounds in exhaust emissions fine particulate matter, we have calculated the ratio
value for Pyrene/BaP ~ 0.7 for noncatalyst-equipped petrol cars, ~1.3 for catalyst-equipped petrol cars
and >16 for heavy duty diesel engines. A recent study from Perrone et al., (2014) reported PM-phased
PAH emission factors for different types of vehicles (Euro 3 standards), the ratio value for Pyrene/BaP
was ~ 6 and 30 for petrol and diesel cars, respectively. The results from both ratios
(Fluorene/Fluorene+Pyrene and Pyrene/BaP) suggest high contribution from petrol engines to particle
composition in Beijing and Delhi (Fig S1).

**3.4 Emission source fingerprints**
The profiles of specific class of PAHs and their abundance vary largely, depending on the fuel types
and combustion conditions (IARC, 2012). Previous studies reported that 2 and 3 aromatic ring PAHs
are mostly emitted from wood combustion (Khalili et al., 1995; Larsen and Baker., 2003; Liu et al.,
2017); 2, 3 and 4 ring from diesel exhaust emission (Bourotte et al., 2005; Ravindra et al., 2007; De
Souza et al., 2016; Zheng et al., 2017); 3 and 4 ring from coal combustion (Harrison, et al., 1996; Liu
et al., 2017); 4, 5, and 6 ring from vehicle emissions (Ravindra et al., 2007, Zhao et al., 2020); 5 and 6
ring from petrol and oil combustion (Harrison, et al., 1996; Ravindra et al., 2007). In this study, we
have classified the 17-PAHs based on their number of aromatic ring; we referred to previous studies
and to our knowledge of local sources, sampling site and surroundings, and analytical uncertainties to
describe the emission source of each class. The 2 and 3 aromatic ring PAHs were below LOD and LOQ
in many samples, however, they are predominantly found in the gas phase and their partitioning to the
particle phase is very small because of their high volatility; their percentage in the particle phase was
previously reported to be less than 10 % (Ravindra et al., 2007; Liu et al., 2017; Zhao et al., 2020).
In this study, the 4, 5, and 6 ring PAHs accounted ~ 95 % of the total $PM_{2.5}$-bound PAHs concentrations
measured in both campaigns (Fig. 6). The mean contribution of the number of ring to the total of PAHs
in Beijing $PM_{2.5}$ was distributed in the order 5 > 4 > 6 ring, and in Delhi as 5 > 6 > 4 ring (Fig. 6). In
both, Beijing and Delhi, road traffic is known to be one of the largest emission source of gas and particle
phase pollutants (Fang et al., 2016; Zhang et al., 2020; Shivani et al., 2019), which might explain the
high contribution from 4, 5 and 6 ring to total PAHs, while diesel and coal combustion may also
contribute to the emission of 4 ring PAHs. The 4 ring PAHs concentration in Delhi was higher than in
Beijing, the mean concentration of the 4 ring PAHs in daytime and night-time samples was 2.7 and 2.1
ng m$^{-3}$ in Beijing; 3.1 and 4.1 ng m$^{-3}$ in Delhi, respectively (Table 1). In contrast to Beijing (5 > 4 > 6
ring), the 4 ring PAHs total concentration in Delhi was lower than the 6 ring PAHs (Fig. 6) and a
potential emission source could be the common use of fuel and oil in power generators.
The distribution of ring PAHs in Figure 6 shows a comparison between the results from this study and
our previous study in winter Beijing (Elzein et al., 2019). The contributions from 2, 3 and 4 rings were
higher in winter than in summer, and the ring PAHs are distributed as 4> 5 > 6 > 2-3 ring (Fig. 6).
Ambient temperature highly affects the gas/particle partitioning of 2-3 ring PAH (Tsapakis and
Stephanou., 2005; Gaga and Ari., 2011; Verma et al., 2017). Therefore, higher contributions from 2, 3
and 4 rings in winter Beijing are likely due to lower temperature and to the use of coal and wood
combustion for residential heating. The 2-3 and 6 ring PAHs contributions in winter Beijing are
relatively similar, while in summer Beijing the contribution from 6 ring PAHs is ~ 20 % higher than 2-
3 ring, most probably due to the effect of ambient temperature on 2-3 ring PAH. The 5 ring PAHs
(representative of vehicle emissions) contribute the most to the total PAH concentration in summer
Beijing and Delhi, suggesting a high contribution from petroleum combustion.
Finally, identifying PAH markers emitted from specific emission sources (types of fuel, types of coal,
types of waste, etc.) in ambient air is still complex due to the similarity of PAH profiles from different



source types, which may quickly blend in the air with interferences from both nearby and remote
emission sources. A more complete assessment of emission source types at specific locations would
require the use of individual PAHs as source markers combined with other chemical constituents of
PM$_{2.5}$ (elements and ions) and with gas phase air pollutants known to be released from the same source
such as VOCs markers.

**3.5 Health risk assessment**
Although PAHs have long been recognized as carcinogenic environmental pollutants, BaP is still the
only PAH allowing a quantitative risk assessment (WHO, 2000, Boström et al., 2002). BaP is used as
the most common reference chemical as being representative for PAH mixtures from emissions of coke
ovens and similar combustion processes in urban air (WHO, 2000). Relative potencies of individual
PAHs (relative to BaP) have also been published as toxicity equivalency factors (TEF) (Nisbet and
LaGoy, 1992; Larsen et al, 1998; Durant et al, 1996, OEHHA., 1994). Thus, the carcinogenic risk of
the mixture of PAHs can be expressed as BaP equivalents (BaP$_{eq}$). The equivalent exposure to the index
compound (i.e. BaP) can be calculated from the TEF of each target compound (Table S2) multiplied by
its corresponding concentration in ng m$^{-3}$.

$$[BaP]_{eq} = \sum_{i=1}^{N} PAH_i \times TEF_i \tag{1}$$

To estimate the statistical potential of contracting cancer from inhalation and lifetime exposure to PM$_{2.5}$-
bound PAHs, commonly known as the lifetime excess cancer risk (LECR) shown in Eq. (2), we have
used the WHO unit risk (UR) estimate of 8.7 x 10$^{-5}$ (ng m$^{-3}$)$^{-1}$ (WHO, 2000), meaning that 8.7 people
per 100 000 people may contract lung cancer when exposed continuously to 1 ng m$^{-3}$ of BaP
concentration over a lifetime of 70 years. This risk refers to the total PAH mixture and not only to the
BaP content (U.S.EPA, 2002, Boström et al., 2002) and is referred to as the surrogate approach. The
use of BaP$_{eq}$ instead of BaP in Eq.2, overestimates the LECR, and therefore the use of the actual
measured BaP concentration better assesses the lifetime cancer risk following Eq. (2)

$$LECR = BaP \times UR \tag{2}$$

**Table 2.** Mean concentration of BaP$_{eq}$, BaP and LECR assessment for Beijing and Delhi.

| Sampling location | Mean [BaP]$_{eq}$ ± SD* (ng m$^{-3}$) | Mean BaP ± SD* (ng m$^{-3}$) | LECR | LECR per million people |
|---|---|---|---|---|
| Beijing / 24 h | 1.47 ± 0.35 | 0.8 ± 0.17 | 7 x 10$^{-5}$ | 70 |
| Delhi / 24 h | 3.42 ± 1.35 | 1.78 ± 0.67 | 15.5 x 10$^{-5}$ | 155 |

* SD : Standard deviation
As shown in Table 2, the LECR attributable to the 15 PAHs in urban air of Beijing and Delhi was 7 x
10$^{-5}$ and 15.5 x 10$^{-5}$ (> 10$^{-6}$), respectively, suggesting an elevated lifetime cancer risk for adults (Chen
and Liao., 2006; Bai et al., 2009), especially when considering the population size associated with each
city. The LECR value for Delhi gives an estimate of 85 additional cancer cases per million people
exposed, in comparison to Beijing. The LECR for Beijing in winter (Elzein et al., 2019) was much
higher than in summer (this study) and showed 1235 additional cases, and this was mostly attributed to
the increase in use of fossil fuels for central and residential heating, in addition to meteorological
conditions such as lower volatilisation at low temperatures and lower photochemical transformation. It
is however the annual mean which is directly related to cancer risk.
Although BaP is widely used as indicator of all PAHs carcinogenicity, this approach is still under debate
and may not give a very good representation of the whole mixture potency (U.S.EPA, 2002, Boström



et al., 2002). Delgado-Saborit et al., (2011) have used the TEFs to calculate the percentage contribution
of each PAH to total carcinogenicity following Eq. (3):

$$(\%\,\text{Carc. Potential})_i = \frac{(RC \times TEF)_i}{\sum_{i=1}^{N}(RC \times TEF)_i} \times 100 \tag{3}$$

where RC is the relative abundance marker of an individual PAH to the carcinogenic marker BaP (RC
= (PAH)$_i$/(BaP)). Using Eq.3, the compounds that contribute most to the total carcinogenic potential of
the PAH mixture in Beijing (B) and Delhi (D) are: Benzo[a]pyrene (B: 46 %; D: 48 %),
Dibenzo[a,h]anthracene (B: 23 %; D: 19 %), Benzo[b]fluoranthene (B: 15 %; D: 13 %),
Benzo[k]fluoranthene (B: 5 %; D: 5 %), and Indeno[1,2,3-cd]pyrene (B: 6 %; D: 10 %). The sum of all
other PAHs used in this study was about 5 %.
Since the majority of people (~ 90%) spend most of their time indoor, the total PAH burden from
inhalation has been related to indoor air, and BaP is used as a marker for the carcinogenic potential of
all PAHs irrespective of the environment (indoor or outdoor) (Delgado-Saborit et al., 2011). In this
study, the health risk evaluation was only based on inhalation exposure to PAHs in the particulate phase.
The risk values can increase due to the presence of PAHs derivatives in the particulate phase such as
the nitrated-PAHs (Elzein et al., 2019), and in particular from 6-Nitrochrysene and 1,6-dinitropyrene
who have been attributed a high TEF value equal to 10 (OEHHA., 1994; WHO., 2003; Lundstedt et al.,
2007). Dermal exposure to PAHs is also an important risk factor for skin cancer but toxicity values for
dermal exposures are still not available (U.S.EPA, 2002), in addition ingestion exposure to PAHs from
soil, sediments and water is high (Li et al., 2010), and both exposures (dermal+ingestion) can highly
exceed the risk from inhalation (U.S.EPA, 2002).
PAHs in the gas phase are mostly low molecular weight (2-3 ring PAHs) and their partitioning to the
particle phase is small (Ravindra et al., 2007; Liu et al., 2017; Zhao et al., 2020) with lower TEF values
(Table S2), therefore, their contribution to total carcinogenic potential is low (< 5 % in this study).
Previous studies may consider a variable number of PAHs and other aromatics with known TEF such
as the nitrated-PAHs, including different references for TEF values which make a direct comparison of
the carcinogenic risk between studies not ideal. However, as particulate-PAH concentrations are lower
in warm months, related BaP$_{eq}$ values were also lower in warm months. Feng et al., 2019 reported an
average BaP$_{eq}$ total concentration of 20 PAHs equal to 1.9 and 21.9 ng m$^{-3}$ in summer and winter
Beijing, respectively. In this study, the summer BaP$_{eq}$ total concentration of 15 quantified PAHs in
Beijing was 1.47 ng m$^{-3}$, while in our previous study for winter Beijing (Elzein et al., 2019), the BaP$_{eq}$
total concentration of 16 quantified PAHs and 7 derivatives was 23.6 ng m$^{-3}$. The results from this study
suggest to focus attention on mitigating the emission of major contributors to the total carcinogenic
potential (Benzo[a]pyrene, Dibenzo[a,h]anthracene, Benzo[b]fluoranthene, Benzo[k]fluoranthene, and
Indeno[1,2,3-cd]pyrene) in order to reduce adverse health effects from exposure to this class of air
pollution. These compounds are 5 and 6 ring PAHs, and were mostly related to emission from petrol
and oil combustion (Harrison, et al., 1996; Ravindra et al., 2007).

**4 Conclusions**
Diurnally-resolved samples of ambient PM$_{2.5}$ were collected in Beijing-China from 22 May 2017 to 10
June 2017 (20 days), and in Delhi from 28 May 2018 to 5 June 2018 (9 days). The 24 h average
concentration of PM$_{2.5}$ was $39 \pm 21$ µg m$^{-3}$ (range: 16 - 97 µg m$^{-3}$) in Beijing, exceeding the Chinese 24
h guideline value (75 µg.m$^{-3}$) on 1 day of the 20 sampling days, while in Delhi the 24 h average
concentration of PM$_{2.5}$ was $143 \pm 27$ µg m$^{-3}$ (range: 110 - 195 µg m$^{-3}$) exceeding the Indian 24 h
guideline value (60 µg.m$^{-3}$), on all 9 sampling days. High contribution to PM$_{2.5}$ emissions was attributed
to residential energy use emissions and to the construction sector.



In Beijing and Delhi, 17-PAHs were quantified using a GC-Q-TOF-MS and the measured
concentrations compared between daytime and night-time, showing a high relative proportion (~ 95 %)
of 4, 5 and 6-ring PAHs in the particle phase. In Beijing, $\sum$17-PAHs concentrations varied between 2.6
and 31.2 ng m$^{-3}$ (average 8.2 ± 5.1 ng m$^{-3}$) in daytime, and from 2.8 to 11.4 ng m$^{-3}$ (average 7.2 ± 2.0
ng m$^{-3}$) at night-time. In Delhi, $\sum$17-PAHs concentrations varied between 8.4 and 36.6 ng m$^{-3}$ (average
13.6 ± 5.9 ng m$^{-3}$) in daytime, and from 13.8 to 42.9 ng m$^{-3}$ (average 22.7 ± 9.4 ng m$^{-3}$) at night-time.
In Beijing, Indeno[1,2,3-cd]pyrene was the highest contributor to the mean total PAHs during daytime
(12 %) and Benzo[b]fluoranthene at night (14 %) at night-time. In Delhi, Indeno[1,2,3-cd]pyrene was
the largest contributor to the total PAHs in both the day (17 %) and night-time (20 %).
The elevated mean concentration of total PAHs in Delhi at night was attributed to emissions from
biomass burning, waste burning, open fire cooking along with meteorological conditions facilitating the
accumulation of air pollutants as a result of low atmospheric boundary layer heights.
The ratio of BaP/BaP+BeP has been used to evaluate the contribution from local sources against long
range atmospheric transport of particle-bound PAHs. This ratio varied between 0.5 and 0.6 in Beijing,
indicating a larger contribution from local sources, while in Delhi, 25 % of the data were below 0.5,
indicating a possible contribution from regional pollution at the sampling site, but local emissions were
still the dominant source of PAHs found in the particle phase. Flu/Flu+Pyr and Pyr/BaP were used as
diagnostic ratios to distinguish between petrol and diesel, and results suggest petrol combustion
emissions as a major source in Both Beijing and Delhi.
PAHs were classified according to their number of aromatic rings to characterize major emission
sources. The 4, 5, and 6 ring PAHs accounted ~ 95 % of the total PM$_{2.5}$-bound PAHs concentrations in
both campaigns. The 5 ring PAHs contribute the most to the total PAH concentration in summer Beijing
and Delhi, suggesting a high contribution from petroleum combustion. In Beijing, the 4 ring PAHs total
concentration was higher than the 6 ring by 8 % during the day and 5 % at night, while in Delhi, the 6
ring PAHs total concentration was higher than the 4 ring PAHs by 7 % during the day and 18 % at
night; a potential emission source of 6 ring PAHs in Delhi could be the common use of fuel and oil in
power generators. Due to the similarity of PAH profiles from different source types, it would be
beneficial to use other source markers such as elements and ions in PM$_{2.5}$ and VOCs markers from the
gas phase to better identify the differences in emission sources.
The lifetime excess lung cancer risk was calculated for Beijing and Delhi, with the highest estimated
risk attributed to Delhi (LECR = 155 per million people), 2.2 times higher than Beijing risk assessment
value (LECR = 70 per million people). The results from this study suggest focusing attention on
mitigating the emission of major contributors to the total carcinogenic potential, being the 5 and 6 ring
PAHs (mostly emitted from petrol and oil combustion), in order to reduce adverse health effects from
inhalation exposure to PAHs in the particulate phase.
Finally, in Beijing, the anti-pollution actions since 2013 appear to have had a positive effect on the air
quality, while in Delhi, despite the government effort to mitigate air pollutants emission, a strict
implementation of emission control policies is still needed with particular focus on mitigating
residential emissions and burning, increasing the electricity supply to cover peak demand in summer
and limiting the use of local power generators as well as, promoting cleaner vehicles. Future studies in
different districts of Beijing (other than Haidian) and Delhi (other than old Delhi) and rural areas would
be helpful for comparison of population exposures and spatial variation. Higher frequency filter
sampling (every 3 h) can provide a better assessment of PAH concentrations and photochemistry and
can lead to better conclusions on direct source emission signals as modified by meteorology during the
daytime and night-time.



*Author contributions:* AE conducted the chemical analysis, analysed the data and prepared the manuscript. ACL, JFH and RMH contributed to the interpretation, writing and corrections of the paper. ERV calculated and provided the data on Delhi planetary boundary layer height. RG supported on site filter collection and helped to set up the laboratory at the field site. AE and SJS conditioned and collected the filter samples in Beijing. GJS and BSN conditioned and collected the filter samples in Delhi. LRC and MSA measured and provided the data on $PM_{2.5}$ in Delhi. All authors reviewed and contributed to corrections of the paper.

*Competing interests.* The authors declare that they have no competing interests.

*Acknowledgements*: Authors gratefully acknowledge the U.K. Natural Environment Research Council for funding Air Pollution and Human Health programme, reference: NE/N007115/1 and NE/N006917/1. We also acknowledge the NERC grant NE/S006648/1. We acknowledge the support from Pingqing Fu, Zifa Wang, Jie Li and Yele Sun from IAP for hosting the APHH-Beijing campaign at IAP. We thank Zongbo Shi, Tuan Vu and Bill Bloss from the University of Birmingham, Siyao Yue, Liangfang Wei, Hong Ren, Qiaorong Xie, Wanyu Zhao, Linjie Li, Ping Li, Shengjie Hou, Qingqing Wang from IAP, Kebin He and Xiaoting Cheng from Tsinghua University, and James Allan from the University of Manchester for providing logistic and scientific support for the field campaigns. Leigh Creilly and Mohammed Alam acknowledge the NERC grant NE/P016499/1. We acknowledge the logistic support of Shivani (Indira Gandhi Delhi Technical University for Women), Eiko Nemitz and Neil Mulligan (Centre for Ecology and Hydrology) and Tuhin Mandal (CSIR-National Physical Laboratory). This work was supported by the Newton-Bhabha fund administered by the UK Natural Environment Research Council, through the Delhi-Flux project of the Atmospheric Pollution and Human Health in an Indian Megacity (APHH-India) programme (grant reference NE/P016502/1). GJS, BSN and SJS acknowledges the NERC SPHERES doctoral training programme for studentships. The meteorological data in this study was taken from National Oceanic and Atmospheric Administration.

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

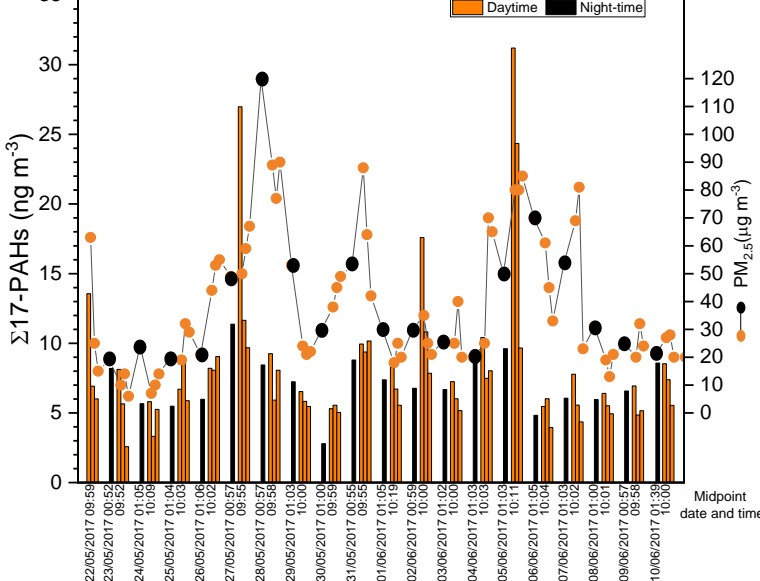


**Figure 1** Temporal variation of total PAHs and PM$_{2.5}$ concentrations in summer Beijing. PM$_{2.5}$ concentrations were averaged to the filter sampling time, approximately 3 h in daytime and 15 h at night. The 3 h midpoint time tick labels at noon (~13:00) and in the afternoon (~16:00) have been omitted for clarity.

1000



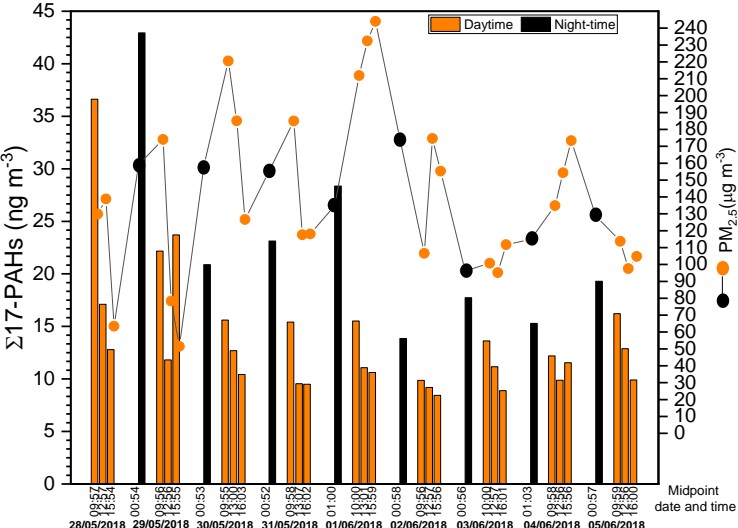

**Figure 2** Temporal variation of total PAHs and PM$_{2.5}$ concentrations in summer Delhi. PM$_{2.5}$ concentrations were averaged to the filter sampling time, approximately 3 h in daytime and 15 h at night.

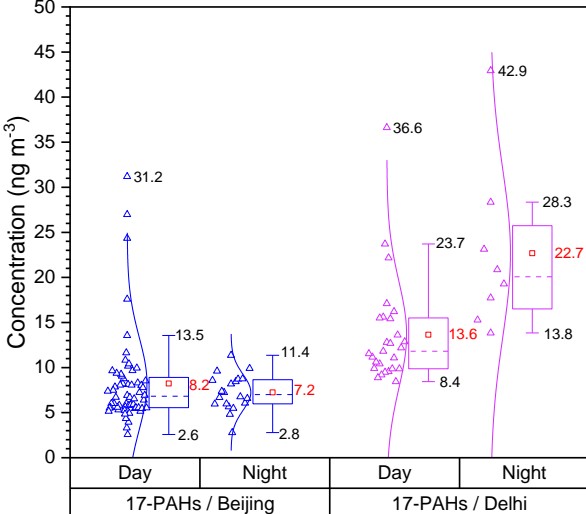

**Figure 3.** Concentrations of 17-PAHs in PM$_{2.5}$ samples during the daytime (3 h) and night-time (15 h). Box plots represents the 25th and 75th percentiles range of the observed concentrations and the whisker numbers reflect the data within 1.5 times the interquartile range (IQR). Red square symbols represent the mean concentration, and the short dash line within the boxes represent the median. Empty Triangles correspond to the data measured over 3 h and 15 h in Beijing (Blue) and in Delhi (Purple). The lines between data points and boxes reflect a normal distribution curve.



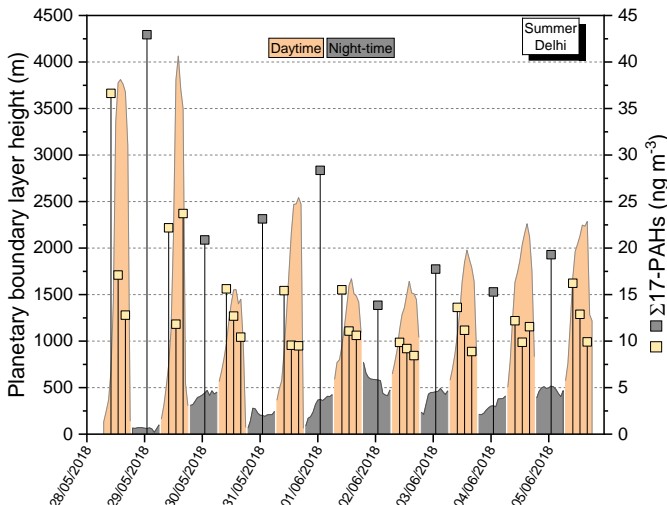

**Figure 4** Planetary boundary layer height in summer Delhi 2018 (Lat 28.625, Lon 77.25; source: ECMWF ERA5 in 0.25°, 1-hour time resolution). Square symbols represent the temporal variation of total PAHs in daytime and night-time.

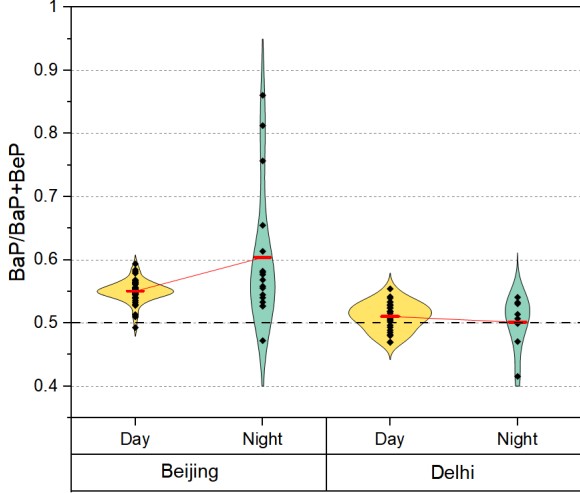

**Figure 5.** Ratio of BaP/BaP+BeP in $PM_{2.5}$ samples during the daytime (every 3 h) and night-time (15 h). Violin plots represent the data distribution between day and night and the wider sections represent the higher density of data smoothed by a kernel density estimator. Red rectangle symbols represent the mean concentration, and the red line connect the mean values between day and night.





**Table 1.** Minimum, maximum and mean concentrations of individual PAHs in PM$_{2.5}$. Compounds in
bold represent the highest mean contribution to the sum of all compounds.

| Compound/ring | Summer Beijing: PAHs Concentrations (ng m$^{-3}$) | | Summer Delhi: PAHs Concentrations (ng m$^{-3}$) | |
| --- | --- | --- | --- | --- |
| | Variation Daytime / Night-time | Mean ± SD* Daytime / Night-time | Variation Daytime / Night-time | Mean ± SD* Daytime / Night-time |
| 17-PAHs | 3 h / 15 h | 3 h / 15 h | 3 h / 15 h | 3 h / 15 h |
| Naphthalene/2 | 0.07-0.29 / 0.02-0.15 | 0.14 ± 0.06 / 0.06 ± 0.04 | 0.07-0.62 / 0.35-0.58 | 0.29 ± 0.14 / 0.45 ± 0.09 |
| Acenaphthylene/3 | 0.08-0.12 / 0.02-0.08 | 0.10 ± 0.03 / 0.04 ± 0.02 | 0.05-0.15 / 0.15-0.29 | 0.09 ± 0.02 / 0.19 ± 0.06 |
| Acenaphthene/3 | < LOD | < LOD | < LOD | < LOD |
| Fluorene/3 | 0.09-0.15 / 0.02-0.07 | 0.13 ± 0.03 / 0.05 ± 0.01 | 0.04-0.09 / 0.03-0.09 | 0.07 ± 0.02 / 0.05 ± 0.02 |
| Phenanthrene/3 | 0.16-1.28 / 0.07-0.42 | 0.37 ± 0.25 / 0.28 ± 0.09 | 0.14-0.58 / 0.56-0.86 | 0.26 ± 0.12 / 0.70 ± 0.10 |
| Anthracene/3 | < LOD | < LOD | < LOD | < LOD |
| Total 2-3 ring PAHs | 0.04-1.30 / 0.06-0.62 | 0.38 ± 0.29 / 0.41 ± 0.13 | 0.15-1.24 / 0.98-1.87 | 0.62 ± 0.28 / 1.36 ± 0.29 |
| Fluoranthene/4 | 0.15-5.78 / 0.25-1.58 | 0.74 ± 0.99 / 0.66 ± 0.31 | 0.36-1.76 / 0.8-1.28 | 0.67 ± 0.29 / 1.04 ± 0.16 |
| Pyrene/4 | 0.12-2.84 / 0.19-1.12 | 0.56 ± 0.58 / 0.56 ± 0.21 | 0.37-2.10 / 0.82-1.36 | 0.69 ± 0.36 / 1.07 ± 0.18 |
| Benzo[a]anthracene/4 | 0.43-1.22 / 0.12-0.53 | 0.56 ± 0.16 / 0.30 ± 0.10 | 0.59-1.59 / 0.49-1.32 | 0.75 ± 0.20 / 0.72 ± 0.27 |
| Chrysene/4 | 0.42-3.27 / 0.28-1.12 | 0.81 ± 0.51 / 0.62 ± 0.20 | 0.64-2.35 / 0.88-1.95 | 0.94 ± 0.34 / 1.20 ± 0.35 |
| Total 4 ring PAHs | 1.14-13.4 / 0.84-4.24 | 2.67 ± 2.19 / 2.14 ± 0.77 | 2.04-7.83 / 3.05-6.17 | 3.10 ± 1.19 / 4.10 ± 1.02 |
| Benzo[b]fluoranthene/5 | 0.44-3.76 / 0.41-1.78 | **0.97 ± 0.65 / 1.02 ± 0.33** | 0.87-3.73 / 1.34-4.86 | **1.40 ± 0.59 / 2.41 ± 1.12** |
| Benzo[k]fluoranthene/5 | 0.56-2.67 / 0.31-1.06 | **0.93 ± 0.43 / 0.68 ± 0.19** | 0.96-3.38 / 1.05-3.44 | **1.39 ± 0.48 / 1.80 ± 0.75** |
| Benzo[a]pyrene/5 | 0.63-2.56 / 0.34-1.15 | **0.95 ± 0.38 / 0.71 ± 0.19** | 0.95-3.41 / 1.10-3.92 | **1.40 ± 0.51 / 2.01 ± 0.90** |
| Benzo[e]pyrene/5 | 0.55-2.08 / 0.12-0.92 | 0.80 ± 0.32 / 0.47 ± 0.22 | 0.88-3.67 / 1.04-5.70 | **1.37 ± 0.61 / 2.14 ± 1.49** |
| Dibenzo[a,h]anthracene/5 | 0.69-0.78 / 0.14-0.26 | 0.74 ± 0.05 / 0.19 ± 0.03 | 0.64-0.91 / 0.12-1.08 | 0.76 ± 0.11 / 0.39 ± 0.27 |
| Total 5 ring PAHs | 0.65-12 / 1.21-4.24 | 3.57 ± 1.97 / 2.98 ± 0.80 | 3.68-15.11 / 4.81-19.0 | 5.69 ± 2.36 / 8.88 ± 4.49 |
| Indeno[1,2,3-cd]pyrene/6 | 0.65-3.27 / 0.39-1.57 | **1.03 ± 0.47 / 0.93 ± 0.31** | 1.32-6.36 / 2.12-8.62 | **2.29 ± 1.11 / 4.45 ± 2.03** |
| Benzo[ghi]perylene/6 | 0.44-2.45 / 0.14-1.37 | 0.77 ± 0.37 / **0.80 ± 0.3** | 1.08-6.08 / 2.01-7.53 | **1.97 ± 1.13 / 3.90 ± 1.76** |
| Total 6 ring PAHs | 0.43-5.71 / 0.66-2.72 | 1.76 ± 0.86 / 1.72 ± 0.54 | 2.47-12.44 / 4.12-15.7 | 4.26 ± 2.24 / 8.34 ± 3.64 |
| Total 17-PAHs | 2.6-31.2 / 2.8-11.4 | 8.2 ± 5.1 / 7.2 ± 2.0 | 8.4-36.6 / 13.8-42.9 | 13.6 ± 5.9 / 22.7 ± 9.4 |
| Total 16-PAHs Winter Beijing (Elzein et al., 2019) | 18-297 / 23-165 | 87.3 ± 58 / 107 ± 51 | | |

* SD: Standard Deviation





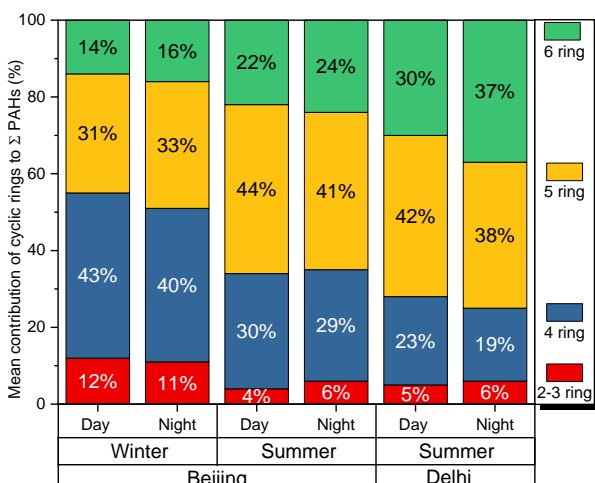

**Figure 6.** Distribution of PAHs compounds by number of cyclic rings in PM$_{2.5}$ samples collected during the daytime (every 3 h) and night-time (15 h) for summer Beijing 2017 and Delhi 2018 (this study) and winter Beijing 2016 (Elzein et al., 2019).