# Peer review of "A comparison of PM2.5-bound polycyclic aromatic hydrocarbons in summer"

_Atmospheric Chemistry and Physics, 2020_

## Referee Comment (RC1) · Anonymous Referee #1 · 30 Aug 2020

Review of "A comparison of PM2.5-bound polycyclic aromatic 1 hydrocarbons in summer Beijing (China) and Delhi (India)" by Elzein et al.

The authors presented measurement results of 17 polycyclic aromatic hydrocarbons (PAHs) in Beijing, China, and Delhi, India in summer. The sampling was conducted with higher time resolutions ( $\sim$  3 hours for daytime samples and  $\sim$ 15 hours for nighttime samples) as compared to traditional 24-h samples. The PAHs were quantified with GC-Q-ToF-MS. Results showed that PAH concentrations were higher in Delhi than those in Beijing, and the summer PAH concentrations were lower than those in winter in Beijing. From the measured PAH profiles, sources of PM-bound PAHs in these two mega-cities

in developing countries were inferred. In addition, health risks were calculated from the measured PAH concentrations. The study is well designed and the analysis is rigorous. The manuscript is fairly well written. I recommend Minor Revision before publication, with a few comments as follows.

**Major**

1. It was stated in the abstract (and in the conclusion) that "in Delhi 25% of the emissions were attributed to long-range atmospheric transport". The only evidence the authors used to support this is on L425-427/P9, showing that 25% of data in Delhi had a Bap/(BaP + BeP) ratio of less than 0.5. This evidence is a little bit thin to support such a statement. I suggest the authors to either elaborate this with more evidence, or tone down such an unsupported statement.

2. The issue of oxidation during sampling to the interpretation of results. First, in the paragraph of L192/P5, the authors noted that this effect could be an additional source of uncertainty (10 - 30%) to conventional analytical uncertainties (25 - 30%). The question is, what is the overall uncertainty if both of these two errors are taken into account? Second, after acknowledging this source of potential negative artifact, the authors used it in Section 3.3 to infer particle aging and then to regional transport of PM. Such inference may be conflicting without quantitative assessment on how such "on-filter" oxidation affect the indicator, i.e., the BaP/(BaP + BeP) ratio. Please clarify.

3. Section 3.5. It is not clear why the authors preferred to use only BaP for the cancer risk calculation. Both Table 2 and L524-527/P12 indicated that other PAHs may contribute another half of the risk. Would the reported LECR per million people values be under-estimated if other PAHs are not taken into account?

**Minor**

1. L146/P4: suggest to change "17-PAHs" to "17 PAHs", and change in a number places (e.g., L223/P5) "24 h mean concentration" to "24-h mean concentration".

2. L156/P4: please change "PAHs concentrations" to "PAH concentrations".

3. L187/P4: why higher error could be attributed to samples analysed previously in wintertime? Memory effect? If so, why were the Delhi samples not affected? Were the Delhi samples analysed after Beijing summer samples?

4. L199/P5: Tsapakis and Stephanou 2003: please use proper citation.

5. L207/P5: please add "that" after "than".

6. L414&L425/P9: please use BaP/(BaP + BeP) consistently.

7. L418/P9: please change "[ratio = 0.5]" to "(ratio = 0.5)".

8. L431-434/P10: this seems like two sentences. Please revise.

9. Figure 3: in addition to the non-preferable "17-PAHs" on the graph and in the caption, I do not see the usefulness of putting "17-PAHs" on the graph. Please remove them on the graph and change to "17 PAHs" in the caption.

10. Figure 5: please change the title of the y axis to "Bap/(BaP + BeP)", as well as that in the caption.

11. Table 1: please change "PAHs concentrations" to "PAH concentrations".

---

## Referee Comment (RC2) · Anonymous Referee #2 · 18 Sep 2020

General comments:

This paper present a comparison of PM2.5-bound PAHs in Beijing and Delhi. The diurnal and nocturnal variations of 17-PAHs in both cities were discussed, then the major emission sources were identified and the health risk was assessed. The novelty, if it is, as the authors state, is the high-time resolution ambient particle samples (every 3 hours during daytime and over 15 hours at nighttime) during the summer season. A natural question to this study would be what causes the PAHs to exhibit the diurnal and nocturnal variation and what is new findings based on the high-time resolution samples. Unfortunately, this paper does not seem to present new findings. The discussions are

also too general. Additionally, the manuscript is generally readable though proofreading and grammar corrections would improve it substantially. Overall, although the data are likely new and can be potentially useful in understanding the variation of PAHs in summer Beijing and Delhi, the paper is not sufficiently rigorous to warrant a publication in ACP.

I use the abstract to illustrate my main concerns of the paper. The abstract says: The mean concentration of particles less than 2.5 microns (PM2.5) observed in Delhi was 3.6 times higher than in Beijing during the measurement period. In Beijing, . . .the highest contribution from Indeno[1,2,3-cd]pyrene, while at night-time . . . the largest contribution from Benzo[b]fluoranthene (14 %). In Delhi, . . . the largest contribution from Indeno[1,2,3-cd]pyrene in both the day and night. Local emission sources were typically identified as the major contributors to total measured PAHs, however, in Delhi 25 % of the emissions were attributed to long-range atmospheric transport. The high contribution of 5 ring PAHs to total PAH concentration in summer Beijing and Delhi suggests a high contribution from petroleum combustion. In Delhi, a high contribution from 6 ring PAHs was observed at night, suggesting a potential emission source from the combustion of fuel and oil in power generators. The lifetime excess lung cancer risk (LECR) was, 2.2 times higher in Delhi (LECR = 155 per million people) than in Beijing risk assessment value (LECR = 70 per million people).

The highlighted results above are not really exciting. In fact, many researches have reported the diurnal and nocturnal variations and source appointment of PAHs in Beijing and Delhi based on the low-time resolution samples. I cannot see new or exciting findings in this paper. So I would suggest they focus on the new findings of the study, which cannot be obtained by the low-time resolution samples.

---

## Author Response (AR1)

Response to referee 1:
Author's response in Blue

The authors presented measurement results of 17 polycyclic aromatic hydrocarbons (PAHs) in Beijing, China, and Delhi, India in summer. The sampling was conducted with higher time resolutions (~3 hours for daytime samples and ~15 hours for nighttime samples) as compared to traditional 24-h samples. The PAHs were quantified with GC-Q- ToF-MS. Results showed that PAH concentrations were higher in Delhi than those in Beijing, and the summer PAH concentrations were lower than those in winter in Beijing. From the measured PAH profiles, sources of PM-bound PAHs in these two mega-cities in developing countries were inferred. In addition, health risks were calculated from the measured PAH concentrations. The study is well designed and the analysis is rigorous. The manuscript is fairly well written. I recommend Minor Revision before publication, with a few comments as follows.
We thank the reviewer for commenting this paper. Your corrections and recommendations helped us to better present the data and we believe the paper has been improved.

**Major**
1. It was stated in the abstract (and in the conclusion) that "in Delhi 25% of the emissions were attributed to long-range atmospheric transport". The only evidence the authors used to support this is on L425-427/P9, showing that 25% of data in Delhi had a Bap/(BaP + BeP) ratio of less than 0.5. This evidence is a little bit thin to support such a statement. I suggest the authors to either elaborate this with more evidence, or tone down such an unsupported statement.
The statement of 25 % was removed from the abstract and conclusions, however, the data from BaP/(BaP + BeP) ratio are still useful to support future studies investigating on local and regional emissions in Delhi. Therefore, we prefer to keep this statement in the text, we reported the results as a possible contribution from regional pollution at the sampling site in Delhi. A new discussion was added to the text and detailed in the next question.

Changes to the text:
L.577-580 removed and replaced by: This ratio suggests a larger contribution from local sources in both cities.

2. The issue of oxidation during sampling to the interpretation of results. First, in the paragraph of L192/P5, the authors noted that this effect could be an additional source of uncertainty (10 – 30%) to conventional analytical uncertainties (25 – 30%). The question is, what is the overall uncertainty if both of these two errors are taken into account? Second, after acknowledging this source of potential negative artifact, the authors used it in Section 3.3 to infer particle aging and then to regional transport of PM. Such inference may be conflicting without quantitative assessment on how such "on-filter" oxidation affect the indicator, i.e., the BaP/(BaP + BeP) ratio. Please clarify.
To determine the bias on the results we have used the ''top-down'' approach where the bias determination can be based on recovery efficiency to correct PAHs concentrations. The analytical uncertainty is due to available information on laboratory test performance. The "on-filter" oxidation is a type of chemical degradation/transformation. Therefore, the values of PAHs concentration has to be considered only as a lower limit due to "on-filter" oxidation. It is, however, different from the analytical uncertainty which estimates the lower and upper limits of the results.
A quantitative assessment of "on-filter" oxidation was not a part of this study, our assessment was based on the concentration of ozone measured at the sampling site and compared to quantitative assessment used in previous studies, as reported in section 2.4 for the error evaluation. BaP and BeP were among the major compounds quantified in this study, and as shown in Table 1, their mean concentrations are similar in the margin of the analytical uncertainties, this support the statement suggesting local emissions as major contributors of PAHs in both cities.

According to Tsapakis and Stephanou (2003) the relative reactivity of BaP, with respect to degradation on glass fibre filters, was 1.6 times higher than BeP. Taking into account the estimated error on sampling artefact in each city and assuming that "on-filter" oxidation was affecting BaP and BeP, as suggested by Tsapakis and Stephanou (2003), the ratio of BaP/(BaP + BeP) will be affected negatively by an average of 4 % (day) and 1.6 % (night) for Beijing, 7 % (day) and 3 % (night) for Delhi.

This assumption will therefore be affecting Delhi results suggesting more contribution from long range transport.

Changes to the text:

L.427: However, this assumption does not take into account the ''on-filter'' oxidation errors during sampling. Tsapakis and Stephanou (2003) reported a relative reactivity of BaP of 1.6 times higher than BeP, with respect to degradation on glass fibre filters. Using the reactivity factor of 1.6, the ratio of BaP/(BaP + BeP) will be affected negatively by an average of 4 % (day) and 1.6 % (night) for Beijing, 7 % (day) and 3 % (night) for Delhi. This assumption will therefore be affecting Delhi results suggesting more contribution from long range transport. Therefore, the indicator of particle aging should be used with careful in the summer season unless ozone ambient concentrations are below 30 ppb, and consequently the negative artefacts are considered not significant (Tsapakis and Stephanou 2003).

L.446: Tsapakis and Stephanou (2003) reported a relative reactivity for BaP, Pyrene and Fluorene of 0.86, 0.82, and 0.68 respectively. The relative reactivity of BaP and Pyrene are similar and therefore does not affect the indicator Pyrene/BaP values. Pyrene is by 20 % more reactive than Fluorene, the ''on-filter'' oxidation has little effect on the indicator Fluorene/Fluorene+Pyrene values, because of the large difference in the defined threshold values which were 6 and 30 for petrol and diesel cars, respectively.

3. Section 3.5. It is not clear why the authors preferred to use only BaP for the cancer risk calculation. Both Table 2 and L524-527/P12 indicated that other PAHs may contribute another half of the risk. Would the reported LECR per million people values be under-estimated if other PAHs are not taken into account?

It is true that other PAHs have high Toxicity Equivalency Factor (TEF) but these TEF are relative to BaP. In Eq. 2 the use of BaP instead of BaPeq was recommended by U.S.EPA, 2002, and Boström et al., 2002, and that's because the unit risk (UR) already include the toxicity values of other PAHs, it is referred to as the surrogate approach.

The use of BaPeq (taken into account other PAHs) instead of BaP will overestimate LECR by 83% for Beijing, and 92 % for Delhi.

**Minor**

1. L146/P4: suggest to change "17-PAHs" to "17 PAHs", and change in a number places (e.g., L223/P5) "24 h mean concentration" to "24-h mean concentration".

"17-PAHs" corrected to "17 PAHs" and 24 h corrected to 24-h in the entire manuscript.

2. L156/P4: please change "PAHs concentrations" to "PAH concentrations".
"PAHs concentrations" corrected to "PAH concentrations" in the entire manuscript.

3. L187/P4: why higher error could be attributed to samples analysed previously in wintertime? Memory effect? If so, why were the Delhi samples not affected? Were the Delhi samples analysed after Beijing summer samples?
In fact, lower %RSD was attributed to samples analysed in wintertime Beijing. The precision of sample replicates in wintertime Beijing and summer Delhi showed better %RSD (<10%) for few compounds as shown in Table S1. However, the maximum %RSD in summer Beijing was 13.7 % which is acceptable. This is a type of random errors where it is difficult to determine the origin of the error. L185 to L187 were removed from the text as this error is not related to lower PAH concentrations in summer Beijing.
Yes, Delhi samples were analysed after Beijing summer samples.

4. L199/P5: Tsapakis and Stephanou 2003: please use proper citation.
Corrected as Tsapakis and Stephanou (2003)

5. L207/P5: please add "that" after "than".
Corrected

6. L414&L425/P9: please use BaP/(BaP + BeP) consistently.
Corrected

7. L418/P9: please change "[ratio = 0.5]" to "(ratio = 0.5)".
 Corrected

8. L431-434/P10: this seems like two sentences. Please revise.
Corrected as follow: We calculated the ratio value for Pyrene/BaP using the data reported in a previous study (Rogge et al., 1993), where the authors quantified more than 100 organic compounds in exhaust emissions fine particulate matter. The ratio value for Pyrene/BaP was ~ 0.7 for noncatalyst-equipped petrol cars, ~1.3 for catalyst-equipped petrol cars and >16 for heavy duty diesel engines.

9. Figure 3: in addition to the non-preferable "17-PAHs" on the graph and in the caption, I do not see the usefulness of putting "17-PAHs" on the graph. Please remove them on the graph and change to "17 PAHs" in the caption.
In Figure 3, $\sum$17 PAHs was added to the y axis and removed from the x axis. 17 PAHs corrected in the caption and in Figure 1 and 2.

10. Figure 5: please change the title of the y axis to "Bap/(BaP + BeP)", as well as that in the caption.
Corrected

11. Table 1: please change "PAHs concentrations" to "PAH concentrations".
Corrected

Response to referee 2:
Author's response in Blue

This paper present a comparison of PM2.5-bound PAHs in Beijing and Delhi. The diurnal and nocturnal variations of 17-PAHs in both cities were discussed, then the major emission sources were identified and the health risk was assessed. The novelty, if it is, as the authors state, is the high-time resolution ambient particle samples (every 3 hours during daytime and over 15 hours at nighttime) during the summer season. A natural question to this study would be what causes the PAHs to exhibit the diurnal and nocturnal variation and what is new findings based on the high-time resolution samples. Unfortunately, this paper does not seem to present new findings. The discussions are also too general. Additionally, the manuscript is generally readable though proofreading and grammar corrections would improve it substantially. Overall, although the data are likely new and can be potentially useful in understanding the variation of PAHs in summer Beijing and Delhi, the paper is not sufficiently rigorous to warrant a publication in ACP. I use the abstract to illustrate my main concerns of the paper. The abstract says: The mean concentration of particles less than 2.5 microns (PM2.5) observed in Delhi was 3.6 times higher than in Beijing during the measurement period. In Beijing, … the highest contribution from Indeno[1,2,3-cd]pyrene, while at night-time… the largest contribution from Benzo[b]fluoranthene (14 %). In Delhi, … the largest contribution from Indeno[1,2,3-cd]pyrene in both the day and night. Local emission sources were typically identified as the major contributors to total measured PAHs, however, in Delhi 25 % of the emissions were attributed to long-range atmospheric transport. The high contribution of 5 ring PAHs to total PAH concentration in summer Beijing and Delhi suggests a high contribution from petroleum combustion. In Delhi, a high contribution from 6 ring PAHs was observed at night, suggesting a potential emission source from the combustion of fuel and oil in power generators. The lifetime excess lung cancer risk (LECR) was, 2.2 times higher in Delhi (LECR = 155 per million people) than in Beijing risk assessment value (LECR = 70 per million people). The highlighted results above are not really exciting. In fact, many researches have reported the diurnal and nocturnal variations and source appointment of PAHs in Beijing and Delhi based on the low-time resolution samples. I cannot see new or exciting findings in this paper. So I would suggest they focus on the new findings of the study, which cannot be obtained by the low-time resolution samples.

We thank the reviewer for commenting this paper and we hope our responses below will better clarify the importance of this manuscript, not only to the scientific community but also to policy makers and local governments. We are sorry that the reviewer didn't find the paper exciting, but rigorous analysis of data and factual interpretation can be high value for species where literature observations are relatively sparse.

This manuscript fits well within one of the science focus of ACP (aerosols, field measurements and chemical composition), and also submitted as part of the wider investigation on Beijing air quality (Special Issue: In-depth study of air pollution sources and processes within Beijing and its surrounding region (APHH-Beijing) (ACP/AMT inter-journal SI)). To the best of our knowledge, there is no published work in ACP showing the day/night variation of PAHs in

Delhi. A direct comparison using the same methods and techniques for PAHs in Beijing and Delhi is also absent from the literature. In this work, we have collected, extracted and analysed particle samples using the same analytical method, which provides a sensitive comparison of PAHs and of emission control policies between two megacities, Beijing and Delhi. This study shows that the adverse health effects from inhalation exposure to PAHs in Delhi is 2.2 times higher than in Beijing, assessed using the same LECR method.

Returning to the question of what causes PAHs to exhibit the diurnal and nocturnal variation, and the new findings from high-time resolution samples. As discussed in section 3.2, seasonal variation plays an important role on ambient concentrations of PAHs. However, the variation between day and night depends on many factors including natural impacts and circumference of the sampling location. In both cities, Beijing and Delhi, the concentration of total PAHs in the first 3 h filter (08:30–11:30) of the day appear to be higher than the rest of the day, suggesting a potential relation with the early morning rush hour time and vehicle emissions. Despite the higher boundary layer height during the day in Delhi, PAHs concentration in the first 3 h filter support their direct emissions from local sources especially petrol vehicles.

The difference in PAHs variation between day and night in Beijing is not significant and emission sources were mostly related to petrol combustion emissions and local emission sources rather than contribution from long range transport. However, in Delhi, PAHs variation between day and night was significant. It could be affected by multiple factors including the spatial variation, the input from multiple local sources, the surroundings of the sampling site, and the meteorological conditions in the north of Delhi. The higher total PAHs concentration at night in Delhi could also be related to emissions from biomass burning, waste burning, solid fuel cooking and heavy duty diesels entering the city at night-time. To the best of our knowledge, this is the first time that there has been an attribution of the higher concentration of PAHs at night time to natural influences such as the lower atmospheric mixing heights as shown in Fig. 4. Another important factor that can negatively impact PAHs concentration in daytime is the higher concentration of ozone and on filter oxidation, discussed in section 2.4, 3.2 and 3.3.

So, the new findings are not just about the high-time resolution samples, but also include the important role of natural impacts on PAHs variation such as the boundary layer height at night-time in Delhi. Moreover, the contribution from long range transport in Delhi need more investigation using correlation studies between different types of pollutants. The new findings identified the major sectors that could be subject to mitigation measures and may improve air quality in both cities. The high-time resolution samples clearly provide a better assessment of PAH concentrations and can reflect direct source emission signals as modified by meteorology during the daytime and night-time.  We note that the reviewer did not include any citations to literature reporting equivalent findings and so we contend that whilst of course previous measurement of PAH have been made in both cities, this paper adds new insight into controlling processes and sources.

**Relevant changes made in the manuscript are marked-up in Blue**
**(L.40; 429-437; 443-445; 450-455; 596-597)**
**Text removed from the manuscript marked up in Red**
**(L.40-41; 187-190; 441-442; 596-600)**

[revised manuscript text omitted]